# The Road to Low-Dose Aspirin Therapy for the Prevention of Preeclampsia Began with the Placenta

**DOI:** 10.3390/ijms22136985

**Published:** 2021-06-29

**Authors:** Scott W. Walsh, Jerome F. Strauss

**Affiliations:** Department of Obstetrics and Gynecology, Virginia Commonwealth University, Richmond, VA 23298, USA; jerome.strauss@vcuhealth.org

**Keywords:** low-dose aspirin, preeclampsia, placenta, eicosanoids, sphingolipids, thromboxane, prostacyclin, isoprostanes, neutrophils, protease-activated receptor 1

## Abstract

The road to low-dose aspirin therapy for the prevention of preeclampsia began in the 1980s with the discovery that there was increased thromboxane and decreased prostacyclin production in placentas of preeclamptic women. At the time, low-dose aspirin therapy was being used to prevent recurrent myocardial infarction and other thrombotic events based on its ability to selectively inhibit thromboxane synthesis without affecting prostacyclin synthesis. With the discovery that thromboxane was increased in preeclamptic women, it was reasonable to evaluate whether low-dose aspirin would be effective for preeclampsia prevention. The first clinical trials were very promising, but then two large multi-center trials dampened enthusiasm until meta-analysis studies showed aspirin was effective, but with caveats. Low-dose aspirin was most effective when started <16 weeks of gestation and at doses >100 mg/day. It was effective in reducing preterm preeclampsia, but not term preeclampsia, and patient compliance and patient weight were important variables. Despite the effectiveness of low-dose aspirin therapy in correcting the placental imbalance between thromboxane and prostacyclin and reducing oxidative stress, some aspirin-treated women still develop preeclampsia. Alterations in placental sphingolipids and hydroxyeicosatetraenoic acids not affected by aspirin, but with biologic actions that could cause preeclampsia, may explain treatment failures. Consideration should be given to aspirin’s effect on neutrophils and pregnancy-specific expression of protease-activated receptor 1, as well as additional mechanisms of action to prevent preeclampsia.

## 1. Introduction

The rationale for low-dose aspirin therapy began in the 1970s with the discovery of thromboxane and prostacyclin [1,2]. Thromboxane is a potent vasoconstrictor and platelet aggregating agent, whereas prostacyclin is a potent vasodilator and inhibitor of platelet aggregation. Both are synthesized from arachidonic acid by action of cyclooxygenase to generate prostaglandin H2, which is then converted by thromboxane synthase to thromboxane or by prostacyclin synthase to prostacyclin. 

In the 1980s, low-dose aspirin was being used to prevent recurrent myocardial infarction and other thrombotic events based on its ability to selectively inhibit thromboxane synthesis without affecting prostacyclin synthesis [3,4,5,6]. The reason this is possible is because the synthesis of thromboxane and prostacyclin is compartmentalized in different cell types. In the systemic circulation, thromboxane is produced by platelets. Platelets do not have nuclei and so cannot regenerate cyclooxygenase when it is inhibited. Therefore, the synthesis of thromboxane is inhibited for the life span of the platelets. Prostacyclin is produced by endothelial cells. Endothelial cells do have nuclei and can regenerate cyclooxygenase, so prostacyclin production is minimally affected by low-dose aspirin. 

## 2. Low-Dose Aspirin for the Prevention of Preeclampsia

Preeclampsia only occurs in the presence of the placenta or placental tissue. Once the placenta is delivered, symptoms clear. Therefore, the placenta is key to understanding preeclampsia, but treatment must correct placental, as well as maternal, abnormalities. In the early 1980s, the placental imbalance between thromboxane and prostacyclin was discovered. The first reports described a decrease in prostacyclin production. Several groups reported a deficiency in prostacyclin in umbilical arteries, uterine vessels, and placental veins in women with preeclampsia [7,8,9]. In 1985, we demonstrated that the reduction in placental prostacyclin was associated with a significant increase in placental production of thromboxane (Figure 1). Normal placentas produced equal amounts of thromboxane and prostacyclin, but in preeclampsia the placenta produced 7 times as much thromboxane as prostacyclin [10]. Other studies later confirmed increased placental production of thromboxane in preeclampsia [11,12,13,14,15,16], with the increase linked to increased phospholipase A_2_ [15], increased cyclooxygenase-2 [16], and increased thromboxane synthase [14] in trophoblast cells. 

With the discovery that there was an imbalance between thromboxane and prostacyclin in preeclampsia, it was reasonable to evaluate whether low-dose aspirin would be effective for preeclampsia prevention. The first clinical trial was published in 1986 by Wallenburg et al. [17]. It was a randomized, placebo-controlled, double-blind trial using 60 mg/day of aspirin. Forty-six normotensive women at 28 weeks’ gestation were judged to be at risk for preeclampsia by increased blood pressure response to infused angiotensin II. Twelve of 23 women taking placebo developed preeclampsia, whereas only 2 of 21 women on aspirin developed preeclampsia. The incidence of preeclampsia was decreased 83% by low-dose aspirin (Figure 2). The investigators concluded that low-dose aspirin may correct the thromboxane/prostacyclin imbalance. 

A plethora of clinical trials followed, reporting varying degrees of effectiveness of aspirin treatment. Two large multicenter intent-to-treat studies were conducted in nulliparous pregnant women given 60 mg/day of aspirin by the NICHD Maternal-Fetal Medicine Unit Network and the Collaborative Low-dose Aspirin Study in Pregnancy (CLASP) trials [18,19]. Only modest decreases in the incidence of preeclampsia were found (Figure 3). The MFM Unit Network study reported no improvement in perinatal morbidity and an increased risk of placental abruption. Interest in low-dose aspirin declined after the MFM Network Unit and CLASP studies due to concerns about placental abruption and small beneficial effect of aspirin. 

However, there were problems with these studies. Regarding placental abruption, only one MFM Network Unit reported this, and abruption was found only on pathologic examination. None were clinically significant, and no other studies previous or since have found an increase in placental abruption due to low-dose aspirin therapy [20]. Another problem was that both the MFM Network and CLASP studies recommended patients use acetaminophen for pain relief. Acetaminophen selectively inhibits prostacyclin without affecting thromboxane [21,22], so the effect of low-dose aspirin to correct the thromboxane/prostacyclin imbalance was compromised. Another major problem was these were intent-to-treat studies. Compliance with low-dose aspirin was not taken into consideration [23]. No drug will work if the patient does not take it. 

Hauth et al. reanalyzed the MFM Network data based on compliance [24,25]. They found that women who were more than 75% compliant in taking their aspirin had a significant decrease in the incidence of preeclampsia, from 5.7% to 2.7%, as well as significant decreases in the incidence of low birth weight, preterm birth, and adverse pregnancy outcomes (Figure 4). Unfortunately, these data were only published in abstract form and did not gain recognition. 

In 2007, the first meta-analysis of low-dose aspirin trials was published by Askie et al., who found that in almost all studies low-dose aspirin reduced the incidence of preeclampsia [26]. Additional meta-analysis studies followed, reinforcing the effectiveness of aspirin. Bujold et al. found that aspirin was more effective when started before 16 weeks [27]. Roberge et al. reported that low-dose aspirin was effective in preventing preterm preeclampsia, but not term preeclampsia [28,29]. These investigators also considered the dose of aspirin. They found that studies that used a dose of aspirin ≥100 mg were more effective in reducing preeclampsia than studies that used a dose <100 mg [29], and Seidler et al. reported a dose response effect for aspirin when comparing studies using ≤81 mg/day to those using >81 mg and up to 150 mg/day [30]. Another study reported that aspirin delays the development of preeclampsia, suggesting this may partly explain why aspirin is more effective in preventing preterm preeclampsia than term preeclampsia because women who would have developed preterm preeclampsia had symptoms delayed to term [31]. The influence of obesity is another factor to consider. A dose of 60 mg/day may have been sufficient in the 1980s when the first clinical trials were started, but since then the United States and other countries have experienced an obesity epidemic. Most study subjects are now overweight or obese, which may explain why meta-analysis studies find that higher doses of aspirin are more effective [29,30,32,33]. 

Overall, the meta-analysis studies demonstrated that low-dose aspirin not only decreases the incidence of preeclampsia, but also preterm birth < 37 weeks, perinatal death, IUGR, and pregnancies with serious adverse outcomes. In 2013 and 2018, the American College of Obstetrics and Gynecology recommended low-dose aspirin therapy for women at risk of preeclampsia, and it is now the standard of care [34,35,36]. 

Consideration should be given to the possibility that the effectiveness of low-dose aspirin could be improved by supplementation with L-arginine, the substrate for nitric oxide synthase. Nitric oxide, like prostacyclin, is a potent vasodilator, so supplementation to increase its production would be beneficial. Supplementation with L-arginine significantly reduced the incidence of preeclampsia in a population at high risk for preeclampsia [37], and a recent study showed favorable effects of L-arginine supplementation in conjunction with low-dose aspirin to improve perinatal outcomes, blood pressure values, and uterine pulsatile index [38]. 

Another consideration is the finding that low-dose aspirin is most effective when started before 16 weeks gestation. This raises the importance of identifying accurate predictive biomarkers for preeclampsia risk to be used in conjunction with maternal characteristics and medical history, so at-risk women can be identified early in their pregnancy and immediately put on low-dose aspirin therapy. 

## 3. Does Low-Dose Aspirin Affect the Placenta?

The actions of low-dose aspirin are generally attributed to selective inhibition of maternal platelet thromboxane; however, beneficial effects must extend to the placenta, which is a major source of eicosanoids. Indeed, preeclampsia only occurs in the presence of placental tissue, and the preeclamptic placenta is characterized by increased thromboxane, decreased prostacyclin, and oxidative stress. Does low-dose aspirin affect the placenta to correct the thromboxane/prostacyclin imbalance and oxidative stress? 

As part of the NICHD Human Placental Project, we undertook a comprehensive evaluation of placental lipids in women with normal pregnancy (NP) and women at risk for preeclampsia who were prescribed aspirin [39]. We found the placenta is a rich source of eicosanoids. We measured 30 eicosanoids in numerous different classes of cyclooxygenase and non-cyclooxygenase metabolites. Ten of these were abnormal in women with severe preterm preeclampsia (SPE). Interestingly, thromboxane (TXB2) was not increased, and prostacyclin (6-keto PGF1a) was not decreased (Figure 5), so the imbalance was not present. However, prostaglandins PGE and PGF were decreased, indicating maternal ingestion of aspirin did affect placental cyclooxygenase. These findings suggest low-dose aspirin therapy corrects the thromboxane/prostacyclin imbalance in the placenta.

Correction of the placental imbalance is possible because thromboxane and prostacyclin are compartmentalized within the placenta (Figure 6). Thromboxane is produced by the trophoblast cells on the maternal side of the placenta, whereas prostacyclin is produced by the placental vasculature on the fetal side [40,41,42]. This allows for selective inhibition of thromboxane because as aspirin enters the maternal intervillous space and starts to cross the placenta, its concentrations are highest in the trophoblast cells to selectively inhibit cyclooxygenase associated with thromboxane production. As aspirin crosses the placenta, its concentration gradually declines according to Fick’s second law of diffusion, sparing prostacyclin production by the endothelial cells of the placental vasculature. Only 34% of aspirin from the maternal side crosses to the fetal side [43]. In vitro studies demonstrated that low-dose aspirin preferentially inhibits placental thromboxane while sparing prostacyclin [43,44,45]. 

We also found evidence that maternal ingestion of aspirin attenuated placental oxidative stress. Two of the most abundant isoprostanes, 8-isoprostane (8-iso PGF2a) and 5-isoprostane (5-iPF2a), which are significantly elevated in placentas of preeclamptic women [46,47], were not elevated in our study of women who developed preeclampsia while on aspirin therapy (Figure 5) [39]. Isoprostanes are accurate markers of endogenous lipid peroxidation. They are prostaglandin-like products formed in vivo by free-radical catalyzed non-enzymatic peroxidation of arachidonic acid [48,49,50]. The finding that two of the most abundant isoprostanes were not elevated in preeclampsia is significant because the placental imbalance between thromboxane and prostacyclin is driven by oxidative stress [25,51]. This may explain why the imbalance was not present. 

The fact that placental isoprostanes did not increase in women taking low-dose aspirin could be due to an indirect effect of cyclooxygenase inhibition. Cyclooxygenase generates reactive oxygen species (ROS) [52], so inhibition of cyclooxygenase could have removed the source of free radicals to generate isoprostanes from arachidonic acid (Figure 7). This idea is consistent with our previous reports that low-dose aspirin inhibits lipid peroxides along with thromboxane in the maternal circulation and in the placenta [43,44,45,53]. This action of aspirin could explain the correction of the thromboxane/prostacyclin imbalance because aspirin removed the driving force. 

Despite aspirin therapy, some women develop preeclampsia. Low-dose aspirin reduces the risk, but it does not prevent the disease in all women. Significant elevations in levels of placental hydroxyeicosatetraenoic acids (HETEs) and sphingolipids with biologic actions that could cause preeclampsia could explain why. 

HETEs are lipoxygenase metabolites of arachidonic acid, and they are, therefore, not affected by low-dose aspirin. The placenta produced four HETEs, two of which, 15-HETE and 20-HETE, were significantly elevated in women who delivered preterm with severe preeclampsia (Figure 8) [39]. Both of these HETEs cause inflammation [54,55,56,57,58,59,60,61], and placental pathologic features of preterm preeclampsia are consistent with chronic inflammation [62]. In addition, 20-HETE promotes hypertension, vasoconstriction, and vascular dysfunction [59,60,61]. Intrauterine production of 20-HETE by the placenta could contribute to reduced uterine blood flow and placental vasoconstriction in preeclampsia, and placental release into the maternal circulation could contribute to maternal hypertension. In this regard, 20-HETE enhances vascular reactivity to angiotensin II. 

Sphingolipids are major constituents of the cell membrane and are involved in cell signaling (Figure 9). They are long-chain fatty acids of various carbon chain lengths that contain a backbone of sphingosine. Sphingolipids include sphingomyelin, ceramide, sphingosine, and sphingosine-1-phosphate. They are involved in inflammatory signaling pathways and implicated in cardiovascular disease [63,64,65,66,67,68]. They are not cyclooxygenase metabolites, and so, are not affected by aspirin. The placenta produced 42 sphingolipids, 5 of which were abnormal in women with severe preeclampsia [39]. All sphingolipids that were abnormal were significantly increased compared to normal pregnancy, including major C:18 forms. D-e-C_18:0_ ceramide, D-e-C_18:0_ sphingomyelin, D-e-sphingosine-1-phosphate (S1P), and D-e-sphinganine-1-phosphate were increased 2-fold to over 4-fold in placentas of women with severe preeclampsia compared to placentas of women with a normal pregnancy (Figure 10). 

Abnormal placental sphingolipid production may contribute to several features of preeclampsia. For example, ceramide induces apoptosis, which may contribute to placental cell death in preeclampsia [69], and S1P inhibits extravillous trophoblast migration [70], and so may contribute to failure of extravillous trophoblasts to effectively remodel the spiral arteries in preeclampsia. S1P is also involved in inflammation, vascular permeability, and the immune response. S1P is an intracellular second messenger, but it is also a blood-borne lipid mediator, and as such, has extracellular actions by binding to S1P receptors. Placental secretion of S1P could be responsible for abnormalities in the maternal circulation. Very little information is available about sphingolipids in pregnancy, but maternal levels of ceramide and S1P have been reported to be elevated in preeclampsia and linked to a placental source [71,72]. 

## 4. Other Considerations Involving Neutrophils and Pregnancy-Specific Expression of Protease-Activated Receptor 1

Normal pregnancy is characterized by leukocytosis caused by proliferation of neutrophils in the 2nd and 3rd trimesters. The number of neutrophils increases 2.5-fold by 30 weeks of gestation in normal pregnancy [73], and the number increases further in preeclampsia [74]. Neutrophils are usually thought of as part of the innate immune system and the first line of defense against infection. A role for neutrophils in non-infectious disease has not been widely considered, but accumulating evidence indicates a role for neutrophils in “sterile” inflammatory diseases [75]. 

For neutrophils to manifest their inflammatory effects, they need to infiltrate tissue, and in women with preeclampsia there is extensive neutrophil infiltration into the maternal systemic blood vessels (Figure 11) [76,77,78,79]. In preeclamptic women, 80–90% of vessels in subcutaneous and omental fat are infiltrated and, although all classes of leukocytes are activated [80,81], vascular infiltration is restricted to neutrophils [77,78]. Neutrophil infiltration is associated with a significant increase in inflammatory markers, e.g., interleukin-8 (IL-8), intercellular adhesion molecule-1 (ICAM-1), cyclooxygenase-2 (COX-2), nuclear factor-kappa B (NF-κB), thromboxane synthase (TBXAS1), and myeloperoxidase (MPO) [76,79,82,83]. The finding of neutrophil infiltration provides a basis for a new way of thinking about vascular dysfunction in preeclampsia. It does not discount the potential role of plasma factors but adds a new dimension to the understanding of the underlying mechanisms of the vascular phenotype. 

### 4.1. Pregnancy-Specific Expression of PAR-1 

Protease-activated receptor 1 (PAR-1), originally known as thrombin receptor, is activated by serine proteases, such as thrombin, neutrophil elastase, and matrix metalloproteinase-1 (MMP-1) [84,85,86]. Activation leads to downstream signaling mechanisms that include the RhoA kinase (ROCK) phosphorylation pathway. ROCK is a recognized mediator of enhanced vascular reactivity, and also regulates the shape and movement of cells. There is pregnancy-specific expression of PAR-1. Wang et al. showed that PAR-1 is expressed on neutrophils, but only during pregnancy [87,88]. This suggests that something associated with the placenta is causing the expression of PAR-1 on circulating neutrophils. 

Figure 12 shows omental fat vessels of preeclamptic and normal pregnant women immunostained for PAR-1. In preeclampsia, PAR-1 is expressed in endothelial cells (EC), vascular smooth muscle (VSM), and in neutrophils flattened and adherent to the endothelium, infiltrated into the vessel, and present in the lumen of the vessel. In normal pregnancy, weak staining is present in the endothelium and neutrophils in the vessel lumen. There is an 8-fold increase in gene and protein expression of PAR-1 in blood vessels of women with preeclampsia [89]. 

### 4.2. PAR-1 Mediates Neutrophil Inflammatory Response in Pregnancy 

Proteases, such as MMP-1, neutrophil elastase, and thrombin, are elevated in women with preeclampsia [90]. The expression of PAR-1 on neutrophils is specific to pregnancy, so its activation by elevated proteases in preeclampsia activates an inflammatory mechanism unique to pregnancy. In normal pregnancy, it makes sense that the expression of inflammatory genes would be silenced. A mechanism for this could be DNA methylation to mask binding sites for inflammatory transcription factors, such a NF-κB. However, if the methylation marks were erased, it would open these sites, possibly leading to increased gene expression. One mechanism for erasing methylation marks involves the recently discovered TET proteins (ten-eleven translocation proteins, aka tet methylcytosine dioxygenases). TET proteins regulate gene expression by enzymatic de-methylation of DNA. They catalyze the conversion of 5-methycytosine (5-mC) to 5-hydroxy-methylcytosine (5-hmC) [91,92,93,94], which is further oxidized and then removed by the DNA base excision repair enzyme, thymine-DNA glycosylase, and replaced with unmodified cytosine [95]. TET enzymes were first discovered in 2009 [93], and little is known about their regulation or role in disease. TET2 is the main TET protein expressed in leukocytes, and its activation has been shown to play an essential role in regulating hematopoietic differentiation, which proceeds in mature cells without cell division normally during emigration from the circulation into tissue [96,97,98]. 

### 4.3. Proteases Activate Neutrophil TET2 and NF-κB to Mediate Inflammatory Response

Protease activation of PAR-1 causes translocation of TET2 from the cytosol into the nucleus in neutrophils obtained from pregnant women as evidenced by immunofluorescence and confocal microscopy (Figure 13) [90]. TET2 (green) is localized to the cytosol in control cells of normal pregnant women. Protease treatment with MMP-1 or elastase results in translocation of TET2 into the nucleus (location identified by DAPI blue) in as early as 15 min, which is consistent for proteins containing a nuclear localization signal (NLS)**.** Nuclear translocation of TET2 coincides with activation of NF-κB. Similar to TET2, protease stimulation of pregnancy neutrophils causes translocation of the p65 subunit of NF-kB (red) from the cytosol to the nucleus. Inhibition of PAR-1, as well as inhibition of ROCK, prevents protease-induced translocation of TET2 and p65 into the nucleus (Figure 13). Inhibition of PAR-1 or ROCK also inhibits inflammatory response as measured by the production of IL-8 and TXB2, which are regulated by NF-κB. Protease treatment of neutrophils from normal pregnant women significantly increases IL-8 and TXB2, demonstrating that proteases stimulate inflammatory response, but when cells are pretreated with PAR-1 or ROCK inhibitors, protease-induced increases in IL-8 and TXB2 are prevented. 

Expression and activation of neutrophil TET2 are increased in preeclampsia. Immunohistochemical staining reveals significantly more staining in omental vessels of preeclamptic women than in omental vessels of normal pregnant women (Figure 14) [90]. In preeclampsia, almost 90% of vessels stain for TET2 with neutrophils infiltrated into the vessel wall, as compared to only 16% of vessels in normal pregnancy with staining. When neutrophils are present in normal vessels, they are usually in the lumen of the vessel. High magnification images reveal dark staining of the polymorphonuclear nuclei of neutrophils in preeclampsia (Panel D), as opposed to diffuse staining in normal pregnancy (Panel C). Nuclear staining suggests TET2 is active in preeclampsia, and activation involves translocation from the cytosol to the nucleus just as observed for TET2 translocation induced by protease activation of PAR-1. Staining for TET2 in preeclamptic vessels mirrors the staining for PAR-1 with staining present in endothelium and vascular smooth muscle (VSM), as well as in neutrophils [90]. This close relationship between PAR-1 and TET2 likely has important implications for vascular inflammation in preeclampsia. 

Figure 15 summarizes the molecular mechanisms for protease activation of pregnancy neutrophils. In normal pregnancy, circulating proteases are not elevated, TET2 and NF-κB are localized to the cytosol, and inflammatory genes are not expressed. In preeclampsia, circulating proteases are elevated and activate neutrophils due to their pregnancy- specific expression of PAR-1. Activation of PAR-1 results in the movement of TET2 and NF-κB from the cytosol to the nucleus and the expression of inflammatory genes. The PAR-1 pathway involves ROCK phosphorylation because inhibition of either PAR-1 or ROCK blocks the movement of TET2 and NF-κB from the cytosol to the nucleus and the inflammatory response. To summarize, elevated levels of proteases in the maternal circulation of preeclamptic women activate neutrophils due to their pregnancy-specific expression of PAR-1. PAR-1 activates ROCK, which phosphorylates TET2 and NF-κB, causing their translocation from the cytosol to the nucleus. The fact that TET2 translocation into the nucleus coincides with movement of NF-κB implicates epigenetic mechanisms and suggests that TET2 may enzymatically de-methylate DNA, opening up transcription factor binding sites for NF-κB, resulting in the expression of inflammatory genes. 

### 4.4. Expression of PAR-1 in the Placenta

Several studies show PAR-1 is expressed in the placenta [99,100,101,102], which is a tissue specific to pregnancy and dysfunctional in preeclampsia. Figure 16 shows staining for PAR-1 in a placental villus. PAR-1 is expressed in the syncytiotrophoblast cells, which are directly bathed by maternal blood. PAR-1 is also present in macrophages of the villous core. 

There is evidence that PAR-1 mediates placental dysfunction in preeclampsia. Because PAR-1 is expressed in the syncytiotrophoblast, elevated levels of proteases in the intervillous space could activate PAR-1, leading to placental dysfunction. For example, protease stimulation of trophoblast PAR-1 causes increased release of the angiogenic factor, sFlt [100,103], by activation of placental NADPH oxidase to generate reactive oxygen species [99]. Activation of NADPH oxidase via PAR-1 could be responsible for placental oxidative stress, which drives the imbalance of increased thromboxane and decreased prostacyclin production [25]. 

A protease activating mechanism of neutrophil and placental PAR-1 could explain why preeclampsia only occurs in pregnant women, and a protease feed-forward scenario could explain why clinical symptoms progressively worsen. Protease activation of PAR-1 could explain other features of preeclampsia. For example, because neutrophils have a limited life span of about 8 days, their rapid turnover would explain why maternal symptoms clear shortly after delivery because new neutrophils not expressing PAR-1 enter the circulation. Some women develop preeclampsia in the immediate post-partum period. Labor is recognized to be an inflammatory process, and even in normal term labor, there is extensive infiltration of neutrophils into maternal systemic vasculature [104]. Women who develop post-partum preeclampsia might have been on the verge of developing preeclampsia, and then neutrophil infiltration with labor pushed them over the edge. 

### 4.5. Central Role for PAR-1 in the Clinical Manifestations of Preeclampsia

Protease activation of PAR-1 may play a central role in the pathology of preeclampsia (Figure 17). Protease activation is involved in the neutrophil TET2 inflammatory response, neutrophil activation, and enhanced vascular reactivity [90,105]. Activation of PAR-1 may explain other pathologic features as well because PAR-1 mediates coagulation abnormalities, platelet aggregation, and thromboxane generation. Protease activation of endothelial PAR-1 activates NF-κB, upregulates cell adhesion molecules (ICAM-1), triggers production of neutrophil chemokines (IL-8), and increases permeability of the endothelium to trigger edema formation [106,107,108,109,110,111,112]. PAR-1 may explain the elevation in angiogenic factors because trophoblast and decidual production of sFlt is stimulated by protease activation of PAR-1 [100,103]. Placental oxidative stress may be explained by protease stimulation of trophoblast PAR-1, which activates NADPH oxidase to generate reactive oxygen species, resulting in the release of sFlt [99]. Activation of NADPH oxidase could also explain the placental imbalance of increased thromboxane and decreased prostacyclin characteristic of preeclampsia because oxidative stress drives this imbalance [25]. The effect of aspirin on PAR-1 signaling should be evaluated. If aspirin interferes with downstream signaling of PAR-1, this would be another action to account for its beneficial effects. 

### 4.6. Placental Activation of Neutrophils

Although all classes of leukocytes are activated in the circulation of women with preeclampsia [80,81,113,114,115], only neutrophils extensively infiltrate maternal blood vessels [76,77,78]. The extensive infiltration of activated neutrophils into blood vessels of women with preeclampsia [76,78,79] could explain systemic vascular inflammation and why multiple organs are affected. The question arises as to how neutrophils are activated. The placenta would seem to be a source for the activator because preeclampsia only occurs in the presence of placental tissue. Lipid peroxides are potent activators of leukocytes [116,117,118], and the human placenta produces lipid peroxides and secretes them into the maternal circulation [13,42,46,119]. In women with preeclampsia, placental production of lipid peroxides is significantly higher than in women with normal pregnancy [13,42,46]. Therefore, it is plausible that activation occurs as neutrophils circulate through the intervillous space and are exposed to lipid peroxides released by the placenta [51,120,121]. 

### 4.7. Inhibition of Neutrophils and Treatment of Preeclampsia with Aspirin

Low-dose aspirin is currently standard of care for the prevention of preeclampsia in high-risk populations. Low-dose aspirin selectively inhibits maternal platelet thromboxane production without affecting prostacyclin production and, as shown above, it appears to also selectively inhibit placental thromboxane production, as well as placental oxidative stress. However, maternal platelets and placental trophoblasts may not be the only aspirin targets. Neutrophils may also be a target. The expression of cyclooxygenase-2 is increased in neutrophils of preeclamptic women [79,122], and aspirin inhibits neutrophil production of thromboxane, as well as the generation of reactive oxygen species [117,118]. Neutrophils could be a major source of thromboxane and oxidative stress due to the marked increase in their numbers during pregnancy. Aspirin treatment might also reduce the infiltration of neutrophils into the maternal blood vessels. Future studies are necessary to address the various mechanisms by which low-dose aspirin is able to reduce the incidence of preeclampsia. 

Low-dose aspirin is currently being used to prevent preeclampsia in women at risk, but given its effectiveness, consideration should be given to the use of aspirin in treating women with preeclampsia. Aspirin was contraindicated for use in pregnancy due to concern that it might reduce amniotic fluid volume or cause closure of the ductus arteriosus. However, this concern may be unwarranted because only 30% of an aspirin dose crosses from the maternal to the fetal side of the placenta [43], and more importantly, the Collaborative Perinatal Project in the 1970s involving over 40,000 pregnant women and their offspring, over 24,000 of whom took aspirin during their pregnancy, 1500 of whom were heavily exposed, found no harmful effects of aspirin use on the neonates [123]. 

## 5. Mysterious Beneficial Effects of Low-Dose Aspirin—Is Cyclooxygenase Involved?

The known mechanism of aspirin is to inhibit cyclooxygenase enzymes, the constitutive COX-1 and the inducible COX-2. However, reports are appearing that aspirin also affects non-cyclooxygenase products. For example, placental soluble fms-like tyrosine kinase 1 (sFlt-1) is elevated in the circulation of women with preeclampsia and implicated in preeclampsia pathology [124,125]. sFlit-1 is not a cyclooxygenase product, but low-dose aspirin reduces hypoxia-induced sFlt-1 release by cytotrophoblast cells in vitro [126,127]. Hypoxia causes oxidative stress and the induction of COX-2, so inhibition of sFlt-1 may be related to aspirin’s ability to decrease ROS generated by COX-2 (Figure 7).

Aspirin has favorable effects through alterations in phosphoproteins, transcription factors, and microRNAs implicated in placental apoptosis and trophoblast migration [128,129,130,131]. Aspirin facilitates trophoblast invasion by regulating a family of microRNAs that inhibit trophoblast invasion [131]. Thus, aspirin may augment extracellular trophoblast remodeling of the spiral arteries, which is deficient in preeclampsia. COX-2 is elevated in the process of apoptosis [129], so aspirin may decrease placental apoptosis by inhibiting COX-2. 

Another puzzling effect of low-dose aspirin is on the regulation of placenta-derived exosomes. Exosomes are lipid bilayer nano-vesicles released by many cells and are present in blood [132]. Their lipid makeup reflects their tissue of origin. Placental exosomes can be specifically identified because they contain microRNAs of the chromosome 19 miRNA cluster that are highly and exclusively expressed by the placenta throughout pregnancy [133,134,135,136]. The placenta releases exosomes throughout pregnancy into maternal blood and placental exosomes are higher in women with preeclampsia and may contribute to endothelial dysfunction [137]. Aspirin has been shown to inhibit exosome formation and shedding by platelets, erythrocytes, monocytes, and vascular smooth muscle cells, and it has been suggested that low-dose aspirin may have a similar beneficial effect on placental exosome shedding and content during pregnancy [137]. 

Most of the studies demonstrating beneficial effects of low-dose aspirin on non-cyclooxygenase products were conducted in vitro or with animal models of preeclampsia. It remains to be shown that these effects occur in pregnant women in vivo. However, these studies expose how much there is to learn about how low-dose aspirin achieves its protective effect. Does aspirin affect non-cyclooxygenase pathways, or are cyclooxygenase metabolites involved in regulating these other pathways? Future studies are needed to address these issues. 

## 6. Conclusions

In summary, low-dose aspirin therapy for the prevention of preeclampsia began with the discovery of an imbalance of thromboxane and prostacyclin production by placentas of women with preeclampsia. Although the benefits of low-dose aspirin are generally attributed to inhibition of maternal platelet thromboxane, they must extend to the placenta. Maternal low-dose aspirin appears to attenuate placental oxidative stress and correct the thromboxane/prostacyclin imbalance. Abnormalities in eicosanoids and sphingolipids not affected by low-dose aspirin may explain why some aspirin-treated women develop preeclampsia. 

Meta-analysis studies provide new considerations for low-dose aspirin therapy beyond those currently recommended by the American College of Obstetrics and Gynecology. These include the following: (1) a higher dose of aspirin of 150 mg/day (or 2 baby aspirin/day) is more effective, (2) aspirin should be started before 16 weeks of gestation, (3) obese women might need a higher dose, (4) low-dose aspirin is most effective in preventing preterm preeclampsia, and (5) compliance is very important and should be emphasized to the patient. 

Neutrophils and the pregnancy-specific expression of PAR-1 also play significant roles in preeclampsia. Proteases are elevated in women with preeclampsia and protease activation of PAR-1 on neutrophils and placental trophoblasts can explain major clinical manifestations of preeclampsia. Additional mechanisms of action of aspirin to prevent preeclampsia should be explored, and consideration should be given to using a standard dose of aspirin and possible supplementation with L-arginine for treatment of women with preeclampsia. 

## Figures and Tables

**Figure 1 ijms-22-06985-f001:**
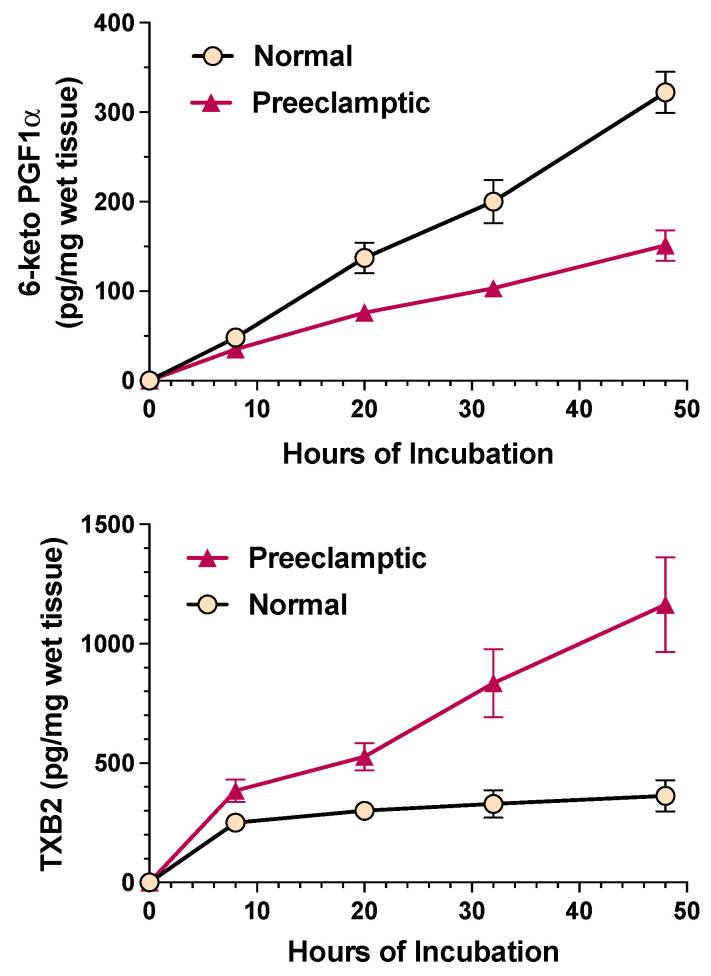
Production of prostacyclin and thromboxane in normal and preeclamptic placentas.

**Figure 2 ijms-22-06985-f002:**
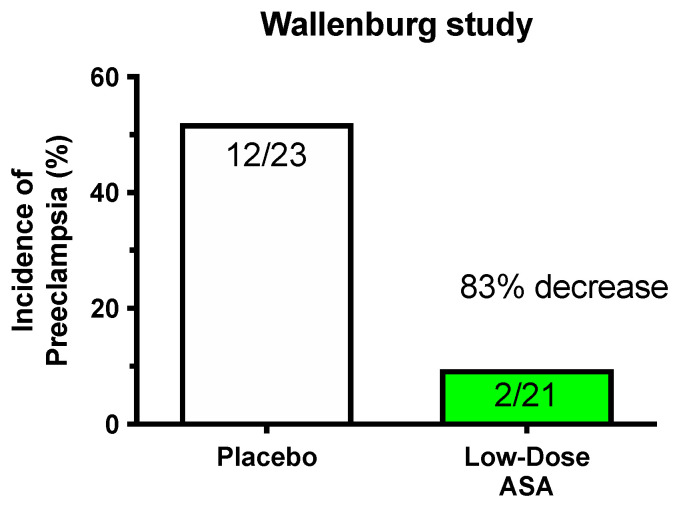
Significant reduction in preeclampsia.

**Figure 3 ijms-22-06985-f003:**
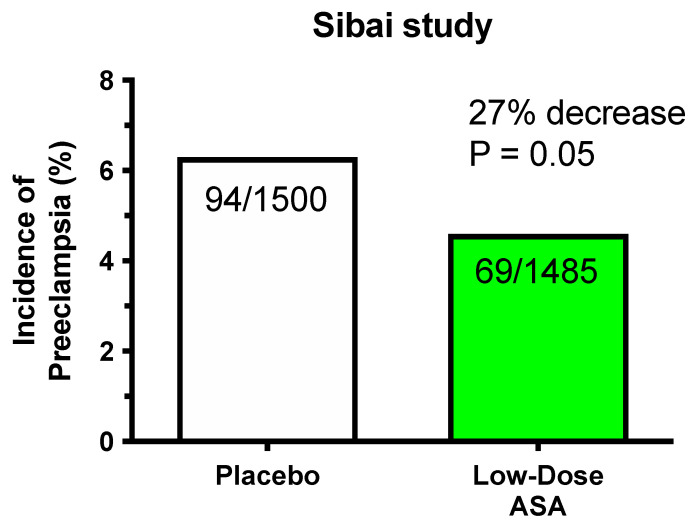
Modest reduction in preeclampsia.

**Figure 4 ijms-22-06985-f004:**
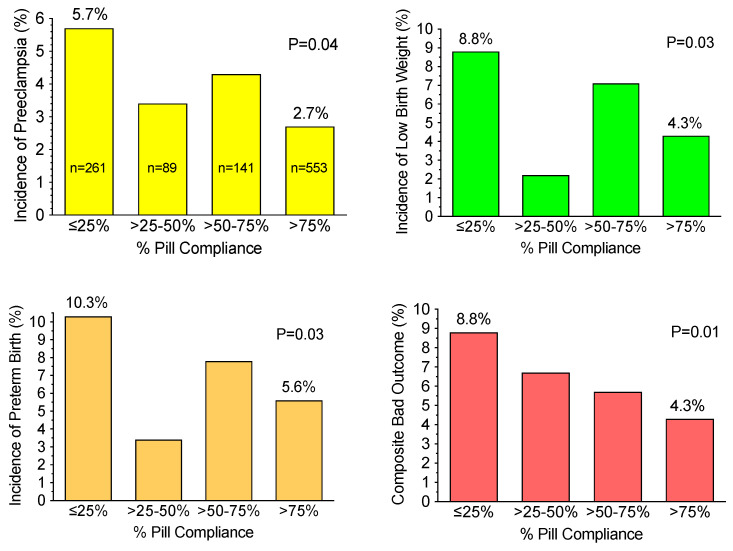
Importance of compliance for low-dose aspirin therapy.

**Figure 5 ijms-22-06985-f005:**
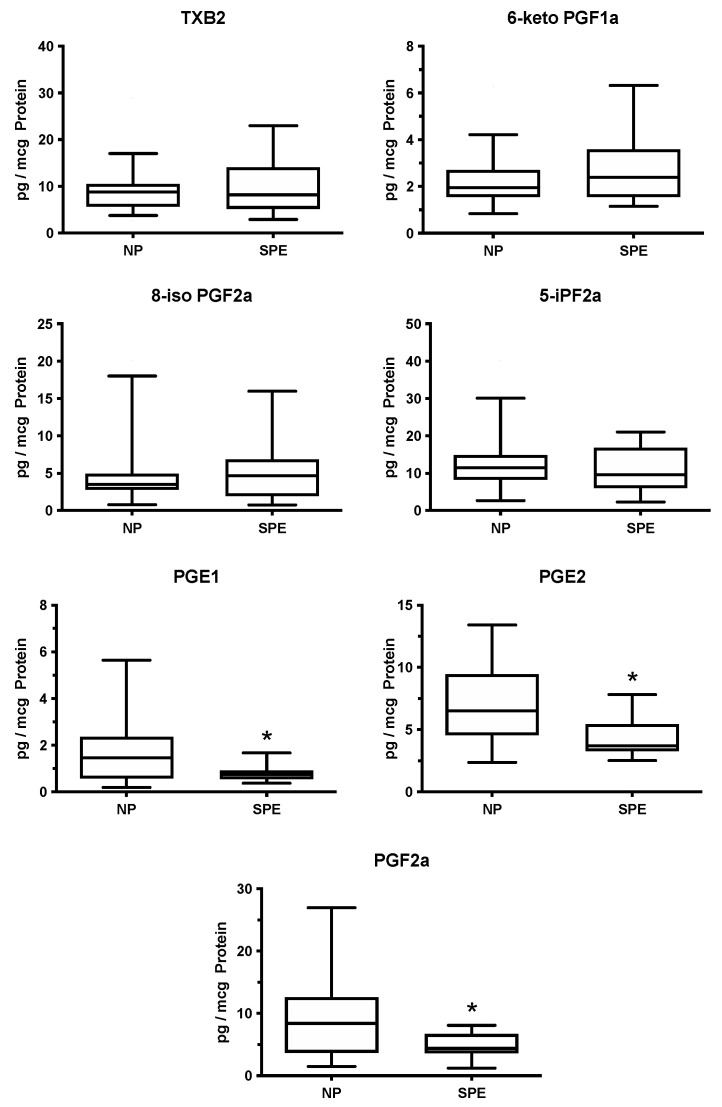
Placental production of cyclooxygenase metabolites and isoprostanes in women with severe preeclampsia receiving low-dose aspirin, * *p* < 0.05.

**Figure 6 ijms-22-06985-f006:**
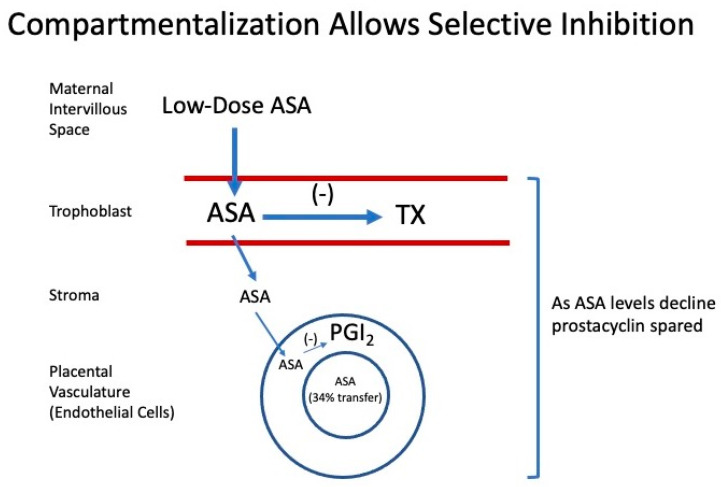
Mechanism for selective inhibition of placental thromboxane (TX) by low-dose aspirin (ASA). Prostacyclin (PGI2).

**Figure 7 ijms-22-06985-f007:**
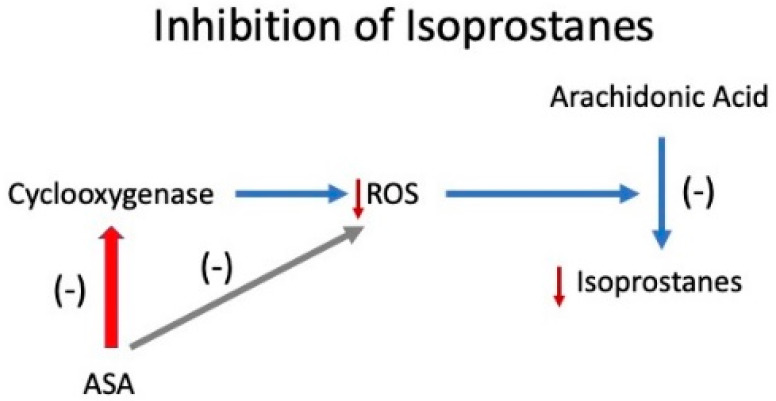
Mechanism for inhibition of isoprostanes by aspirin (ASA).

**Figure 8 ijms-22-06985-f008:**
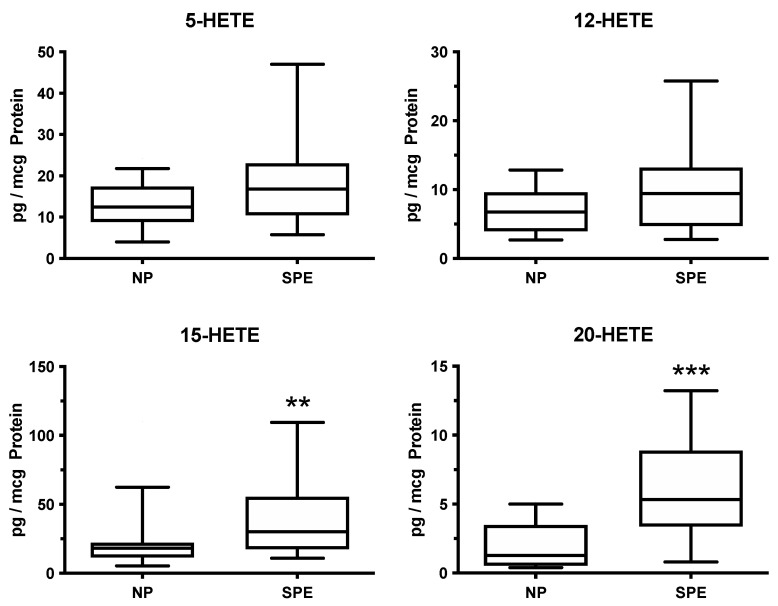
Increases in HETEs related to the development of preeclampsia, ** *p* < 0.01, *** *p* < 0.001.

**Figure 9 ijms-22-06985-f009:**
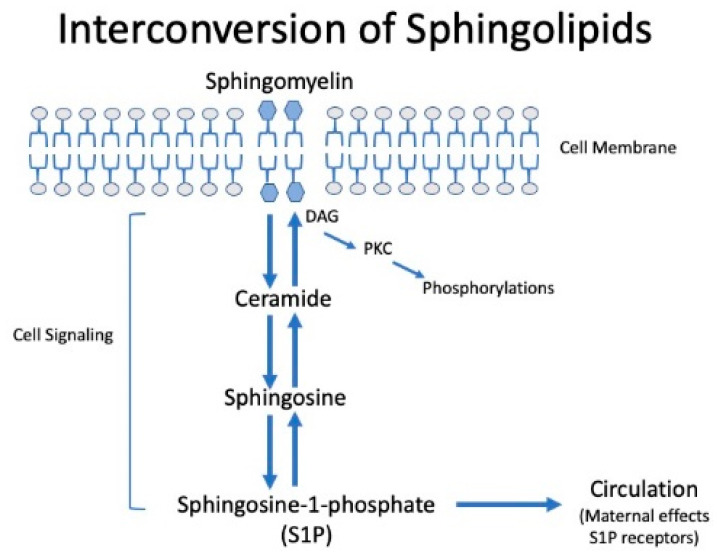
Interconversion of sphingolipids. (DAG, diacylglycerol; PKC, protein kinase C).

**Figure 10 ijms-22-06985-f010:**
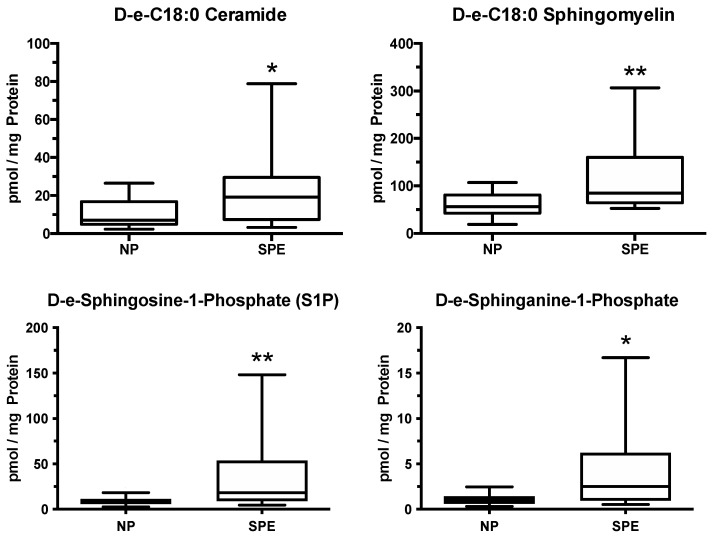
Increases in sphingolipids related to the development of preeclampsia, * *p* < 0.05, ** *p* < 0.01.

**Figure 11 ijms-22-06985-f011:**
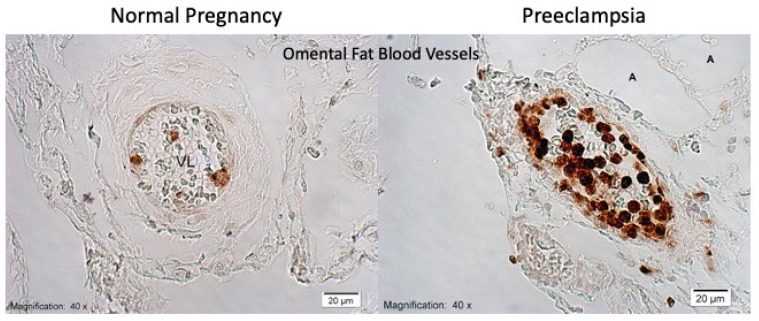
Neutrophils (brown) in omental fat arteries.

**Figure 12 ijms-22-06985-f012:**
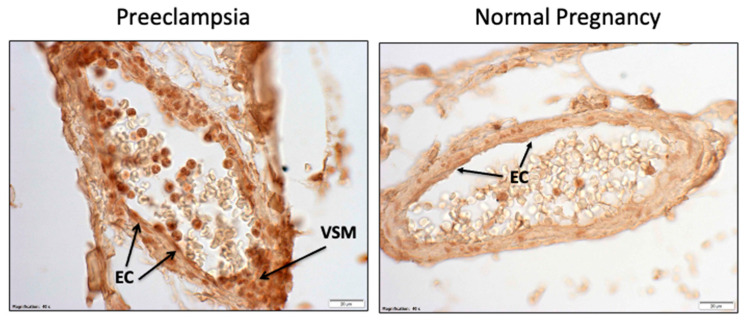
Expression of PAR-1 in omental vessels.

**Figure 13 ijms-22-06985-f013:**
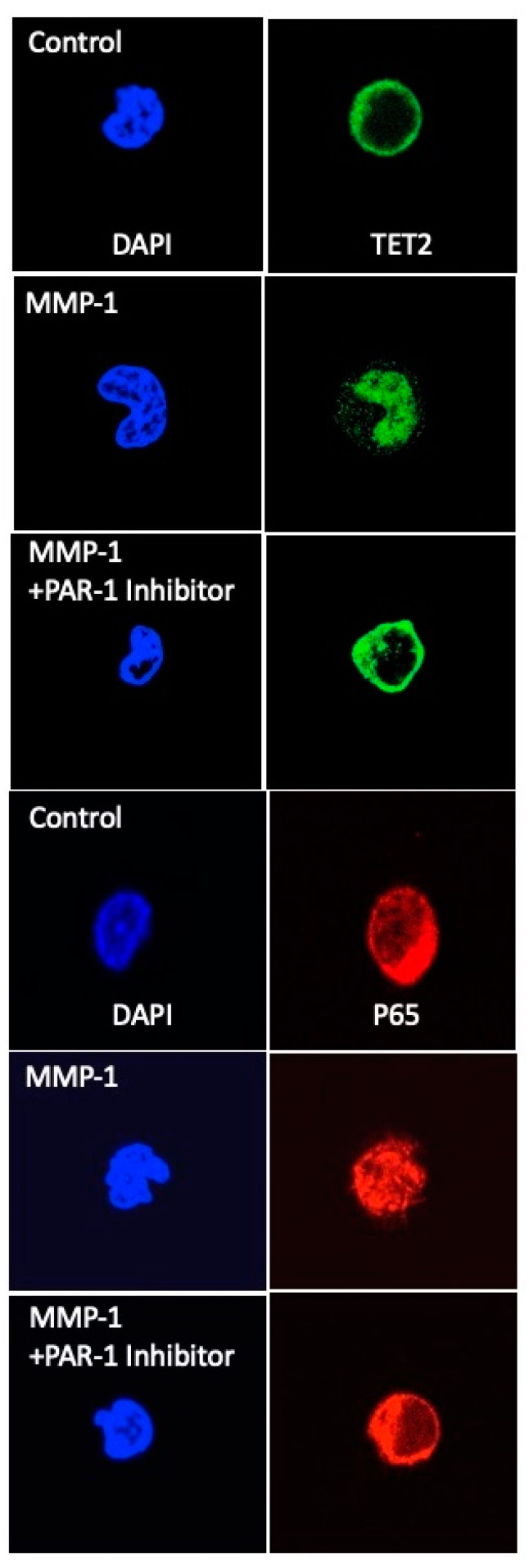
Confocal images of TET2 and p65 immuno-fluorescence staining in neutrophils of normal pregnant women. Nuclear localization induced by protease treatment was prevented by inhibition of PAR-1. Images were taken with a Zeiss LSM 700 using x63 lens and then cropped.

**Figure 14 ijms-22-06985-f014:**
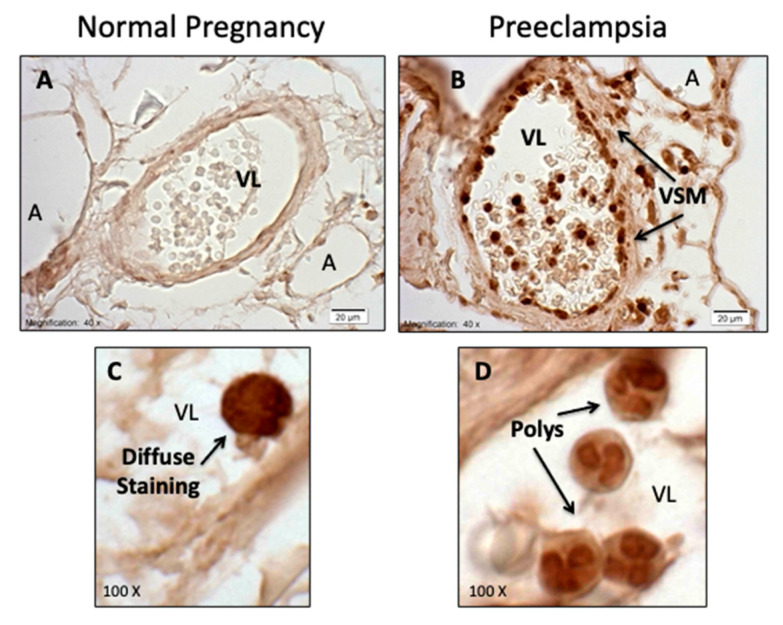
TET2 in omental vessels (**A**–**D**).

**Figure 15 ijms-22-06985-f015:**
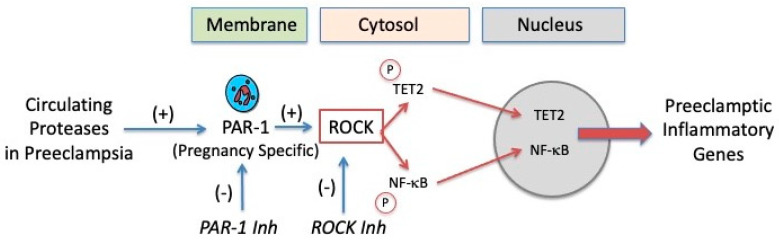
Molecular mechanism for protease activation of pregnancy neutrophils.

**Figure 16 ijms-22-06985-f016:**
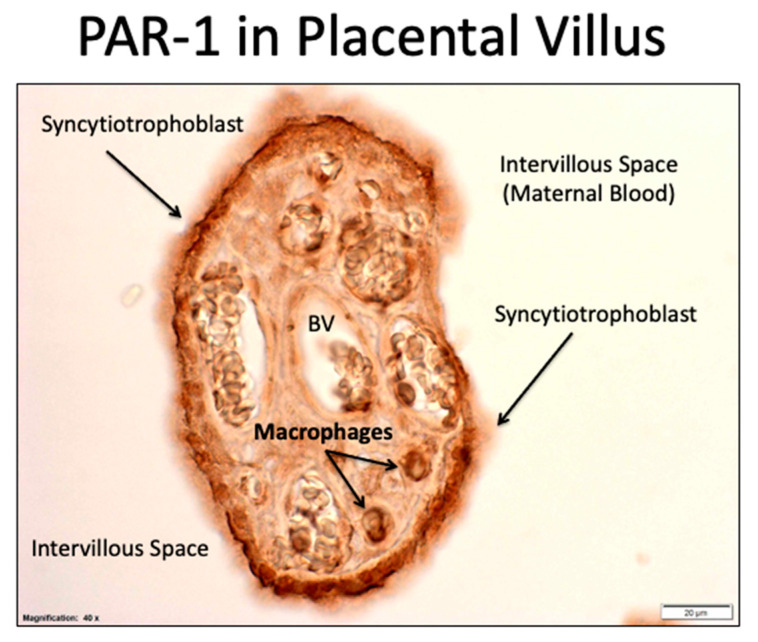
Expression of PAR-1 in syncytiotrophoblasts and macrophages in the placenta (dark brown staining).

**Figure 17 ijms-22-06985-f017:**
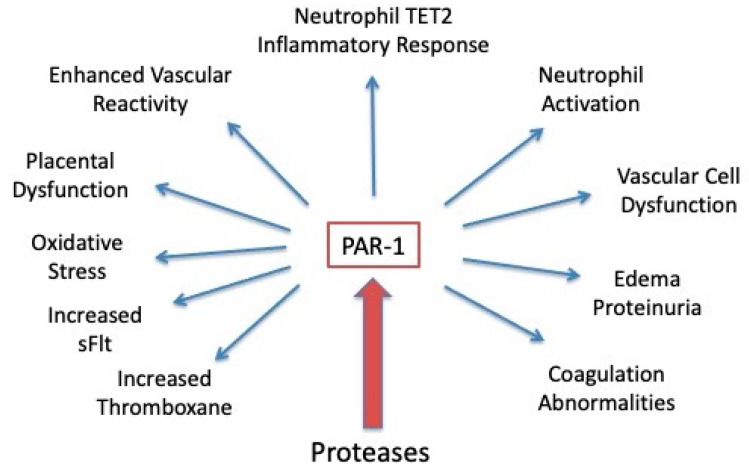
Clinical manifestations of preeclampsia that can be explained by protease activation of PAR-1.

## Data Availability

No new data were created or analyzed in this study. Data sharing is not applicable to this article.

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
