# Peer review of "The Road to Low-Dose Aspirin Therapy for the Prevention of Preeclampsia Began with the Placenta"

_ijms, 2021, doi:10.3390/ijms22136985_

Round 1

Reviewer 1 Report

                This is an interesting review of the way by which low-doses of aspirin protects against preeclampsia through a specific action on the placenta. This is quite new, since, surprisingly the way by which this protection is achieved has not really be systematically studied.

The paper is largely based on some specific axes modulated in the placenta under aspirin treatment, in connection with results obtained by the authors of the paper, which explain the number of illustrations originating from data obtained by the authors, namely, on thromboxane/prostacyclin imbalance, prostaglandin synthesis related to inflammation, inhibition of oxidative stress and of isoprostanes, sphingolipid conversion, neutrophil content in connection with the expression of PAR-1, and their action on TET2 and NFKB.

In sum, the authors emphasize a central function of the protease PAR-1 in the modulation of preeclampsia effects.

                Overall, I think that the data provided are interesting but hardly comprehensive about the effects of aspirin on preeclampsia in the placenta. I am not convinced that the pathways presented as affected are the whole explanation of aspirin effect. I would suggest the incorporation in this review of numerous articles dealing directly with the effects of aspirin in placental function.

I would suggest that the authors propose an extended version of their article where they disucss and integrate recent data on aspirin function in the preeclamptic patients and their placenta. For instance, they should add the recent data on arginin supplementation (PMID: 33978350), the function on trophoblast invasion (PMID: 33865015), the effects on the phosphoproteome (PMID: 32774705), the regulation of STOX1 expression by aspirin (PMID: 32018223) the links with extracellular vesicles regulation (PMID: 31492014) the effects on sFLT1 (PMID: 31335509, 31004367), the putative role on coagulation and complement genes (PMID: 31098302), and probably other aspects.

                This would allow to have a more balanced study, where the pleiotropic effects of aspirin will be presented in their protective function.

Author Response

Response to Reviewer’s Comments:

We thank the reviewers for their very favorable and positive comments and their help in improving our manuscript.

Reviewer 1.

We incorporated the articles suggested by the reviewer dealing with additional favorable effects of low-dose aspirin that have recently been reported for non-cyclooxygenase products. A new section “Mysterious beneficial effects of low-dose aspirin – Is cyclooxygenase involved?” was added as Section 5 (Lines 438 – 481)), and statements regarding L-arginine supplementation were added at the end of Section 2 (Lines 127 – 137). We agree that the inclusion of these studies makes the review more balanced and thank the reviewer for making this suggestion and providing references.

Reviewer 2 Report

Thank you very much for allowing me to review this manuscript.

In my opinion, the Authors raise a very important topic and relevant issue. I recommend this manuscript for publication.

I have no competing interests.

The authors present a very interesting approach to a well-known topic. They discuss not only the existing recommendations for prophylaxis of preeclampsia with acetylsalicylic acid, but also discuss possible reasons for the ineffectiveness of this prevention of preeclampsia. Moreover, the authors' observations regarding the possible endpoints  or targets of aspirin's actions are very interesting.

It is well done manuscript. Very interesting discussion. Properly selected references. Concise and correct conclusions.

A few specific concerns:

  • It seems that optimal strategies of prophylactic use of aspirin boil down to the most accurate identification of high - risk preeclampsia population a little earlier than before the 16th week of pregnancy, taking into account possible inaccurate dating of pregnancy.
  • To maximize effectiveness of prophylaxis with aspirin it should be given  in the evening or at night.

Author Response

Response to Reviewer’s Comments:

We thank the reviewers for their very favorable and positive comments and their help in improving our manuscript.

Reviewer 2.

A statement was added to the end of Section 2 indicating the importance of better strategies, such as accurate biomarkers, for identifying at-risk women before 16 weeks gestation (Lines 138 – 142).

We agree with the reviewer that to maximize prophylaxis with aspirin, circadian rhythms should be considered.

Round 2

Reviewer 1 Report

The addition of specific paragraphs is wellcome. Of course, the paper did not changed drastically in terms of its organization, but at least some topics outside the main research of the authors is presented and do not mislead the reader towards a too monolithic vision of aspirin fucnction. 

While this may not be as well constructed that what could be expected, I think that as is this review is timely and original.